# Relationship between Cycling Infrastructure and Transportation Cycling in a Small Urban Area

**Richard Larouche** [1,*] , **Nimesh Patel** [1] **and Jennifer L. Copeland** [2]

1   Faculty of Health Sciences, University of Lethbridge, 4401 University Drive,
    Lethbridge, AB T1K 3M4, Canada; nimesh.patel@uleth.ca
2   Department of Kinesiology and Physical Education, Faculty of Arts and Science, University of Lethbridge,
    4401 University Drive, Lethbridge, AB T1K 3M4, Canada; jennifer.copeland@uleth.ca
*   Correspondence: richard.larouche@uleth.ca; Tel.: +1-403-332-4439

**Abstract:** The role of infrastructure in encouraging transportation cycling in smaller cities with a low prevalence of cycling remains unclear. To investigate the relationship between the presence of infrastructure and transportation cycling in a small city (Lethbridge, AB, Canada), we interviewed 246 adults along a recently-constructed bicycle boulevard and two comparison streets with no recent changes in cycling infrastructure. One comparison street had a separate multi-use path and the other had no cycling infrastructure. Questions addressed time spent cycling in the past week and 2 years prior and potential socio-demographic and psychosocial correlates of cycling, including safety concerns. Finally, we asked participants what could be done to make cycling safer and more attractive. We examined predictors of cycling using gender-stratified generalized linear models. Women interviewed along the street with a separate path reported cycling more than women on the other streets. A more favorable attitude towards cycling and greater habit strength were associated with more cycling in both men and women. Qualitative data revealed generally positive views about the bicycle boulevard, a need for education about sharing the road and for better cycling infrastructure in general. Our results suggest that, even in smaller cities, cycling infrastructure may encourage cycling, especially among women.

**Keywords:** bicycling; infrastructure; gender; road safety; attitudes; habits; mixed-methods research

## 1. Introduction

It is estimated that the transportation sector is responsible for about a quarter of global greenhouse gas emissions [1]. Moreover, emissions attributable to the transportation sector have increased by 29% between 2000 and 2016 [2]. A growing body of evidence suggests that replacing motorized trips by active transportation (e.g., walking and cycling) is a promising climate change mitigation strategy that can generate considerable health co-benefits [1,3–6]. For example, a multi-site longitudinal study in Europe indicated that if an average person replaced a single car trip/day by cycling for 200 days/year, their mobility-related $CO_2$ emissions would decline by about 0.5 tonnes/year [3].

In addition to the environmental benefits, consistent evidence shows that active transportation is associated with increased physical activity [7–9]. Cycling for transportation is also associated with improved cardiovascular fitness and reduced cardiovascular risk factors [7,10]. Two prospective cohort studies reported that it can reduce the risk of premature mortality by around 40% [11,12]. Despite the numerous potential benefits, only 1.4% of Canadian commuters reported cycling as their main travel mode to work according to data from the 2011 National Household Survey, and this proportion decreased with age [13]. Thus, we need more evidence to understand what interventions and infrastructure can effectively increase rates of cycling to promote sustainability and population health.

Cities, regions, and countries that have been successful at promoting cycling have typically implemented a package of interventions [14–17]. This can make it difficult to

disentangle which type of infrastructure is most effective [17]. Nevertheless, studies of cyclists' or potential cyclists' preferences can provide some insight [18,19]. In the metropolitan Vancouver area, both current and potential cyclists stated that they preferred cycling along residential streets with traffic calming (e.g., speed humps, narrower intersections, etc.) rather than roads with painted bicycle lanes [19]. Using global positioning systems, Broach and colleagues [18] found that cyclists in Portland (Oregon), especially women, went out of their way to use bicycle boulevards to a greater extent than to use the more common painted bicycle lanes.

Bicycle boulevards are streets with low traffic volumes where an array of measures is implemented to reduce the speed of vehicles (e.g., traffic calming, signs and barriers for motorists, roundabouts, etc.) to create safer routes for cyclists [20]. Stop signs in the direction of bicycle travel can also be removed to provide a faster and more direct routes [14], which can make travel time by bicycle more competitive with car travel time [21]. Studies examining the effects of bicycle boulevards on cycling have been conducted in larger cities such as Portland, Oregon and Vancouver, British Columbia [18,19,22,23], and most found them to be preferred by cyclists and/or associated with increases in cycling. In contrast, studies examining infrastructure preferences in Brisbane and Melbourne, Australia, found that women preferred off-road paths over on-road bicycle lanes, a category that can include bicycle boulevards [24,25].

Although larger metropolitan cities in North America have invested in cycling infrastructure for some time [17], this is a newer phenomenon in many smaller cities. Therefore, the potential of bicycle boulevards and off-road paths to increase cycling in smaller cities with a low prevalence of active transportation remain largely unknown. When smaller communities implement new cycling infrastructure for the first time, it is important to examine how it is perceived by community members. Given that the effects of built environment attributes such as walking and cycling infrastructure can be context-specific [26,27], natural experiments have recently been recommended as an approach to examine the effects of local policies and programs [26].

Therefore, the purpose of this study was to examine the correlates of self-reported cycling for transportation in a small city that had recently introduced new cycling infrastructure. Our second objective was to explore if participants reported any change in their cycling behaviour over the 2-year period when the new infrastructure was built. Our third objective was to examine how residents perceived the new cycling infrastructure and any suggestions to make cycling safer and more attractive.

## 2. Materials and Methods

### 2.1. Setting and Study Design

Lethbridge, Alberta is a small urban centre (population~100,000) in western Canada where, according to the 2016 Canadian census, only 1.4% of commuters cycled to work [28]. The city has a well-established network of multi-use paths that are conducive to leisure activities such as cycling, walking, and running [29]. In 2017, the City adopted a 20-year Cycling Master Plan and one of the first deliverables was the construction of a 2.4-km bicycle boulevard on 7th Avenue South, which is a central location. Developed with input from the community, including residents of the targeted neighbourhoods, the bicycle boulevard project involved reducing the speed limit from 50 to 30 km/h, reducing the number of stop signs, and installing mini-roundabouts, traffic diverters, and new traffic control lights [29]. Several negative media stories about the bicycle boulevard were published in the first two years following construction. They notably focused on the issue of drivers ignoring traffic diversion measures, including speeding and driving on the sidewalk [30], safety concerns associated with intersection modifications [31], and construction costs [32,33]. These stories made the case for the present study more compelling.

We investigated the reported travel behaviours and potential correlates of cycling among people intercepted on streets with and without cycling infrastructure. In addition to the bicycle boulevard, we worked with the municipal Transportation Engineering

department to identify two control streets that shared similar traffic volume and speed limits in 2016 and where there had been no recent changes in cycling infrastructure. One of the control streets (6th Avenue North) had no cycling infrastructure and the other (20th Avenue South) had a segregated multi-use path that was constructed over five years prior to the bicycle boulevard. Figure 1 illustrates the location of the bicycle boulevard and control streets. We used an embedded mixed-methods design wherein qualitative data supplemented a predominantly quantitative study [34].

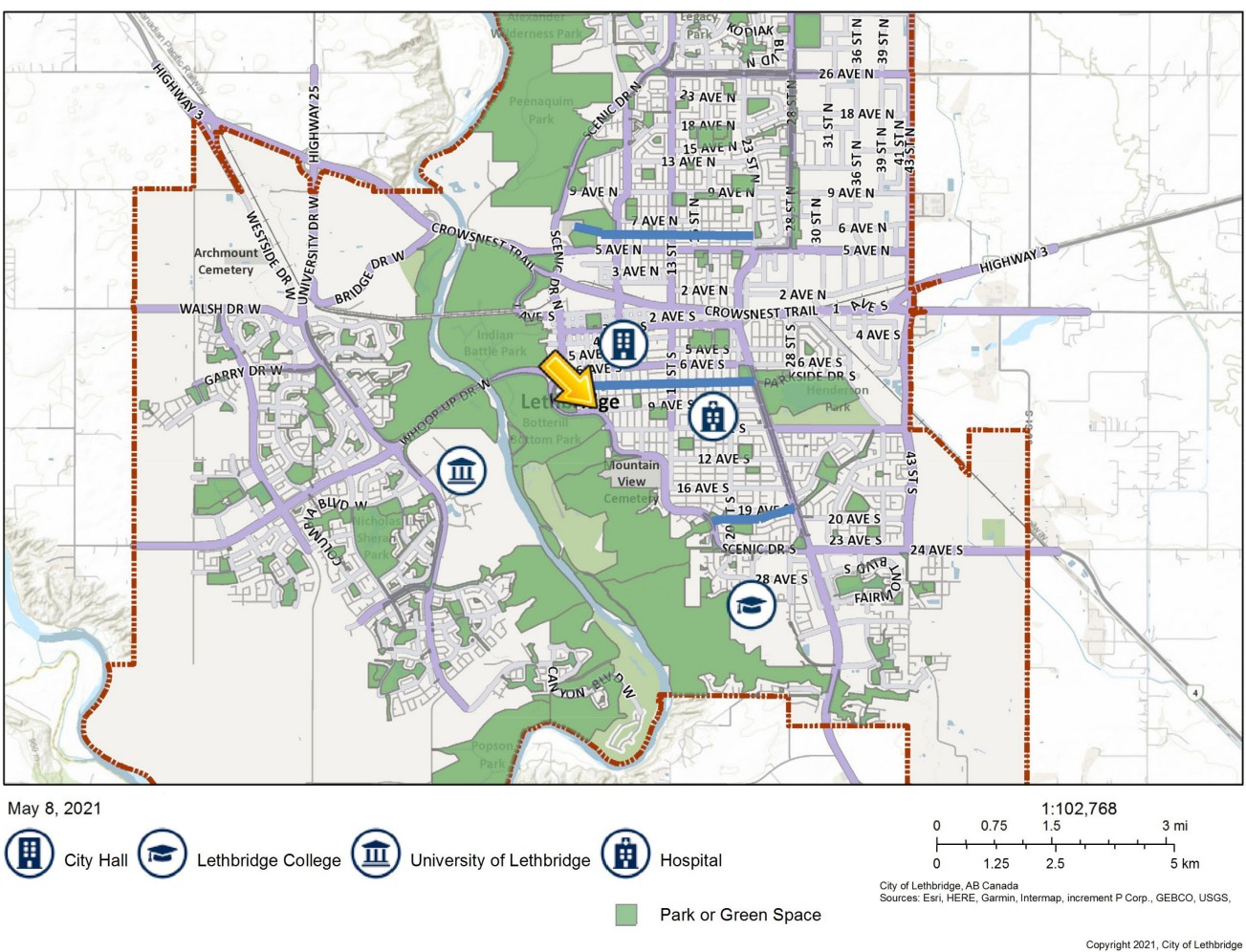

**Figure 1.** Overview of the three streets included in the Cycling in Lethbridge study. (Note: Streets are shown in blue above the map. The street with no cycling infrastructure (6th Avenue North) is at the top of the image, the bicycle boulevard (7th Avenue South) is in the middle, and the street with a segregated bicycle path (20th Avenue South) is at the bottom. The yellow arrow illustrates a connection that was missing at the time of data collection between the bicycle boulevard and a north–south segregated path that runs along Scenic Drive, a major arterial road.)

### 2.2. Participants

Trained research assistants approached individuals traveling along the bicycle boulevard and control streets near intersections and at parks located on these streets. To minimize selection bias, research assistants visited each data collection site in a rotating basis and invited all adult-looking passersby to participate, regardless of the activity that they were doing when intercepted (except those using motorized vehicles, for safety reasons). Data collection occurred at various times throughout the day (between 7:00 a.m. and 9:00 p.m.) between May and August 2018 to avoid restricting our sample to commuters working at

regular business hours. We used a University banner to draw attention to the study, and all staff wore University T-shirts or sweaters. Inclusion criteria were being ≥18 years of age and able to answer questions in English. Individuals who met these criteria were eligible whether they cycled or not. No information could be collected from individuals who refused to participate. The study was fully anonymous, approved by the institutional research ethics committee, and conducted in accordance with the Declaration of Helsinki. All participants provided verbal informed consent and were eligible for a draw to win a $100 gift certificate at a local bicycle shop. We recruited a convenience sample of 246 participants (45.9% women) along either the bicycle boulevard (*n* = 130), a control street without cycling infrastructure (*n* = 50), or a street with a multi-use segregated path (*n* = 66). We attempted to recruit approximately half of participants on the bicycle boulevard as no changes in cycling were expected on the control streets. A sample size calculation performed with the G*Power 3.1.9.7 software indicated that, with the 3-group design, 225 participants were needed to achieve a power of 0.80 to detect a moderate effect size (Cohen's f = 0.25) at an $\alpha$ of 0.05. Figure 2 illustrates the number of participants interviewed at each location, and the number of participants eligible for quantitative and qualitative analyses (described below).

**Interviews**
- *n* = 246 interviews completed
- Bicycle boulevard: 130
- Control street without cycling infrastructure: 50
- Control street with a segregated path: 66

**Quantitative analyses**
- *n* = 243 participants eligible for gender-stratified models*
- Bicycle boulevard: 127
- Control street without cycling infrastructure: 50
- Control street with a segregated path: 66

**Qualitative analyses**
- *n* = 179 participants responded to the qualitative question
- Bicycle boulevard: 98
- Control street with no cycling infrastructure: 37
- Control street with a segregated path: 44

**Figure 2.** Flow of participants in the study. (Note: * Two participants did not report their gender and one did not identify as man or woman; they were excluded from gender-stratified analyses).

### 2.3. Measures

We asked participants to verbally respond to questions listed in Supplementary File S1. Research assistants recorded answers on paper or on a tablet. Participants were provided the following preamble, which did not specifically mention the bicycle boulevard: "We are interested to learn more about how you travel from place to place, and the things that affect your choice of method of transportation." We assessed cycling, walking, and car travel time using International Physical Activity Questionnaire (IPAQ) items [35]. These items were used to assess travel behaviour at the time of the interview (2018) and two years prior (2016; before the construction of the bicycle boulevard). The IPAQ questionnaire has shown good test–retest reliability (Spearman's rho ≈ 0.80) and its convergent validity with accelerometer scores is comparable to other questionnaires [36]. We assessed attitudes towards cycling, walking, and driving with scales developed by Cao and colleagues [37],

which had satisfactory internal consistency in a previous study of bicycle boulevards in Portland, Oregon ($\alpha$ = 0.66–0.84) [23]. Higher scores on these scales indicate more positive attitudes towards a given travel mode. We assessed cycling habit strength with the Self-Report Behavioural Automaticity Index, a 4-item scale that has demonstrated good reliability and predictive validity [38]. We adapted three items about road safety concerns from the Neighborhood Environment Walkability Scale [39,40]. Higher scores on these items indicated greater safety concerns. We adapted three items assessing the subjective norm for cycling from a study by Lemieux & Godin [41]. The survey also included questions about socio-demographic variables that have been shown to influence cycling including gender, age, level of education, and ownership of motor vehicles and bicycles [7,42–44].

### 2.4. Qualitative Data and Analyses

At the end of the survey, participants were asked an open-ended question: "What else do you think might be needed to make cycling safer and more attractive in the City of Lethbridge?" Answers were audio-recorded or hand-written depending on the participants' preference. Audio-recorded responses were transcribed verbatim. Qualitative data were analyzed based on the Framework Analysis strategy [45], which was specifically developed for applied policy analysis. This process consists of five iterative steps: (1) familiarization with the data; (2) identification of a thematic framework; (3) indexing (e.g., coding) of interview transcripts; (4) charting; and (5) mapping and interpretation. The lead author and two research assistants constructed the thematic framework based on emerging themes. Then, two research assistants independently coded the participants' answers. Any disagreement was resolved in discussion with the lead author and decisions were made by consensus. Following Swallow et al. [46], we used a Microsoft Excel spreadsheet to compile coded extracts and the themes to whom they were assigned. In our summary of findings, we included "deviant" cases to illustrate the diversity of views expressed by participants.

### 2.5. Quantitative Data Treatment and Analyses

We computed scales for attitudes towards cycling, walking, and driving, cycling habit, subjective norms, and perceived safety by averaging individual items, then calculated Cronbach's alpha. We truncated reported time spent cycling and walking at 1260 min/week (i.e., 3 h/day) as recommended by the IPAQ developers [47]. We did a series of t-tests and Kruskal–Wallis tests to examine differences between the three streets in socio-demographic characteristics, potential correlates of cycling, and time spent cycling, walking, and driving.

To examine the correlates of cycling, we used generalized linear models based on a negative binomial distribution given the overdispersion of reported cycling time. Informed by previous research [24,25] and preliminary analyses suggesting that the association between cycling infrastructure and time spent cycling differed by gender ($p_{[interaction]}$ = 0.034; Appendix A), we present models stratified by gender. Following a process similar to Gropp et al. [27], we first examined the association between all potential correlates and cycling time in models that controlled only for the street where participants were interviewed. Variables associated with cycling time at the $p < 0.2$ threshold were retained and included in a multivariable model. Then, we used a backward selection process to remove variables with $p > 0.05$, with the exception of street, age, and level of education, which were deemed mandatory variables. Prior to running models, we centred continuous predictor variables at the grand mean and dummy-coded categorical predictors with >2 levels. We used the Akaike Information Criteria (AIC) as indicator for model fit. To explore if reported cycling time differed between 2016 and 2018, we used a one-way analysis of variance. All analyses were performed with IBM SPSS, version 26 (Armonk, NY, USA).

### 3. Results

#### 3.1. Descriptive Statistics

Socio-demographic characteristics of the sample stratified by interview location are provided in Table 1. Overall, participants' age varied from 18 to 100 years (mean = 48.2 ± 18.3) and 45.9% of participants were women. Participants surveyed along the street without cycling infrastructure were younger, less educated, and less likely to own bicycles and motor vehicles (all $p < 0.05$). Based on data from the 2016 census, both the bicycle boulevard and the street with no cycling infrastructure were located in census tracts that had lower household income relative to the Lethbridge census metropolitan area ($75,452/year). The street with the segregated bicycle path was in an average income neighbourhood. Descriptive statistics indicated no significant differences between streets in reported cycling, walking, and driving time (all $p > 0.05$). In general, participants had a positive attitude towards walking and cycling and perceived good subjective norm for cycling. Participants surveyed along the street without cycling infrastructure were more concerned about road safety in their neighbourhood in 2016 and 2018 ($p = 0.001$). Reported cycling time increased from 155 ± 263 to 211 ± 323 min/week between 2016 and 2018 (F = 4.05; $p = 0.045$), but changes in cycling time did not differ by street. In a generalized linear model that included only gender as a predictor of cycling time, men reported cycling 2.40 times more than women (95% CI = 2.00–2.89; $p < 0.001$).

**Table 1.** Sociodemographic characteristics of the sample stratified by interview location.

| Variable | *n* | Bicycle Boulevard (*n* = 130) | Separated Multi-Use Path (*n* = 66) | No Bicycle Infrastructure (*n* = 50) | Test Statistic | Cronbach α |
|---|---|---|---|---|---|---|
| Gender (% women) | 244 | 43.3 | 50.0 | 48.0 | 1.751 | N/A |
| Vehicle ownership (% yes) | 242 | 73.2 | 89.4 | 53.1 | 38.197 *** | N/A |
| Bicycle ownership (% yes) | 242 | 84.2 | 86.4 | 66.7 | 16.299 *** | N/A |
| Education (% ≤ high school) | 241 | 25.4 | 9.1 | 30.7 | 28.714 *** | N/A |
| Age (years ± SD) | 238 | 48.9 (17.3) | 47.1 (16.8) | 43.0 (21.9) | 8.827 * | N/A |
| Cycling time (min/week ± SD) | 245 | 206 (315) | 211 (293) | 222 (383) | 1.537 | N/A |
| Walking time (min/week ± SD) | 240 | 317 (338) | 259 (282) | 297 (309) | 1.145 | N/A |
| Driving time (min/week ± SD) | 242 | 252 (334) | 226 (214) | 441 (684) | 1.393 | N/A |
| Cycling attitude (mean ± SD) | 239 | 4.1 (1.0) | 4.1 (1.0) | 3.8 (1.0) | 3.146 | 0.77 |
| Walking attitude (mean ± SD) | 240 | 4.0 (1.0) | 3.9 (1.1) | 4.0 (1.0) | 0.101 | 0.73 |
| Driving attitude (mean ± SD) | 239 | 2.9 (0.9) | 2.7 (1.0) | 3.0 (0.8) | 4.035 | 0.55 |
| Subjective norm (mean ± SD) | 237 | 3.9 (1.1) | 3.8 (1.1) | 3.8 (1.1) | 1.636 | 0.81 |
| Cycling habit strength (mean ± SD) | 238 | 3.2 (2.2) | 2.9 (2.1) | 3.0 (2.2) | 0.741 | 0.96 |
| Perceived neighbourhood safety concerns (mean ± SD) | 236 | 2.4 (0.8) | 2.2 (0.9) | 2.8 (0.8) | 13.153 ** | 0.84 |
| Median household income of census tract ($–2016 census) [a] | N/A | 56,299 | 73,452 | 55,522 | N/A | N/A |

Note: SD: standard deviation. Test statistic are values of chi-squared for categorical variables and Kruskal–Wallis test statistic for continuous variables. * indicate statistical significance ($p < 0.05$); ** indicates $p < 0.01$; *** indicates $p < 0.001$. [a] median household income is based on weighted data from the 2016 Canadian census for the census tract(s) that correspond to the interview locations.

#### 3.2. Correlates of Time Spent Cycling

Gender-stratified multivariable models of the correlates of cycling time are provided in Table 2. In these models, exponentiated regression coefficients [Exp(β)] represent the average percent change in the outcome variable for each unit increase in independent variables. For example, women surveyed along the street with a segregated multi-use path reported 2.35 times more cycling time compared to those surveyed on the street with no cycling infrastructure (Exp(β) = 2.35; 95% CI = 1.29–4.28). Lower age, higher education levels, greater habit strength, and having a more favorable attitude towards cycling were also associated with more cycling. In contrast, having a more favorable attitude towards walking was associated with less cycling time.

Men surveyed along the bicycle boulevard reported lower cycling time compared to those surveyed on the street with no cycling infrastructure. Higher age, more favorable attitudes towards cycling, greater habit strength, and higher road safety concerns were associated with higher cycling time. Men with college education cycled less than those with lower education levels. Having more favorable attitude towards walking was associated with lower cycling time. Bicycle and vehicle ownership, attitudes towards driving, and perceived subjective norms were not independently associated with cycling time in women or men.

**Table 2.** Correlates of reported cycling time stratified by gender.

| Variable | Women | | Men | |
|---|---|---|---|---|
| | Exp(β) | 95% CI | Exp(β) | 95% CI |
| Street—segregated path (ref: no infrastructure) | 2.35 | 1.29–4.28 ** | 0.54 | 0.29–1.01 |
| Street—bicycle boulevard (ref: no infrastructure) | 1.11 | 0.59–2.10 | 0.44 | 0.25–0.76 ** |
| Education—university (ref: ≤ high school) | 3.00 | 1.64–5.51 *** | 1.00 | 0.60–1.65 |
| Education—college (ref: ≤ high school) | 4.16 | 1.94–8.20 *** | 0.52 | 0.32–0.86 * |
| Age (year) | 0.98 | 0.96–0.99 ** | 1.02 | 1.01–1.03 ** |
| Attitude toward cycling (each unit increase) | 1.97 | 1.48–2.62 *** | 2.29 | 1.49–3.52 *** |
| Attitude toward walking (each unit increase) | 0.34 | 0.24–0.47 *** | 0.63 | 0.50–0.79 *** |
| Cycling habit strength (each unit increase) | 2.19 | 1.64–2.93 *** | 1.85 | 1.38–2.47 *** |
| Perceived road safety (each unit increase) | - | - | 1.84 | 1.44–2.34 *** |

Note: * indicate statistical significance ($p < 0.05$); ** indicates $p < 0.01$; *** indicates $p < 0.001$. Akaike Information Criteria for women model: 1149.818. Akaike Information Criteria for men model: 1512.717.

### 3.3. Qualitative Results

A total of 179 participants responded to the qualitative question, and Table 3 summarizes the main themes and subthemes that emerged. Five main themes were identified: (1) cycling infrastructure; (2) road infrastructure (not specific to cycling); (3) promotion and education; (4) policy; and (5) public perceptions and attitudes. Some responses were assigned to an "other" category (*n* = 19), which included 8 responses suggesting that cycling is good or safe enough in Lethbridge. The most frequent subthemes were issues about sharing the road (*n* = 68), "unspecified routes" (e.g., participants mentioned the need for more bicycle routes in general terms; *n* = 59), issues about cycling safety in general (*n* = 40), the 7th Avenue bicycle boulevard (*n* = 30), and the need for better connectivity in cycling infrastructure (*n* = 22). To represent the type of infrastructure that participants mentioned, the concept of dedicated routes was divided into on-street (including bicycle boulevards), off-street, protected, and unspecified routes. Many respondents were emphatic about the need for physical separation from traffic whereas others emphasized the need for on-road facilities for transportation cycling. Altogether, the need for more dedicated cycling routes was mentioned by 102 participants. Fifteen percent of men recommended more on-street infrastructure compared to 6% of women. In contrast, women were more likely to speak about issues with cycling safety in general (29% vs. 16%).

Themes were often interconnected in participants' responses. For instance, a 60-year-old woman who did not report cycling for transportation mentioned: *"I live very near the intersection of 9th street and 7th avenue [i.e., the bicycle boulevard], and people at the diversion were not stopping at all. So, people in the neighbourhood informed the city and the police that this was a serious problem, and somebody was going to get hit. They made the sign bigger, they have ticketed a lot of people, so people are very aware."* This comment relates to the themes of the bicycle boulevard, safety, signage, and enforcement. A 34-year-old man who reported using a mix of cycling, walking and driving for transportation mentioned: *"Well I love 7th Avenue, I always come here when I am biking. For me, if there were other routes to use, separated bike lane on 13th or on 6th, something like that. That would be very helpful. In general, I find that motorists are not very educated, they stop and don't know what to do about me. I wish that they would understand that I just follow the rules of the road like they do. So, a bit of education in terms*

*of drivers would be helpful. And also, it's June and there is still a bunch of gravel on the side of the road. It's really annoying and needs to be cleaned.*" This participant's comments relate to the themes of the bicycle boulevard, protected bike lanes, connectivity, sharing the road, and maintenance. Moreover, the subtheme of sharing the road includes many comments about the need for education targeted at motorists and/or cyclists. It was often mentioned jointly with the subtheme of safety and/or the need for dedicated cycling infrastructure.

**Table 3.** Themes and subthemes mentioned by the participants to make cycling safer and more attractive in Lethbridge.

| Theme | Subtheme | Total Frequency, $n = 179$ (%) | Frequency in Women, $n = 86$ (%) | Frequency in Men, $n = 93$ (%) |
|---|---|---|---|---|
| Cycling infrastructure | On-street dedicated routes | 19 (11) | 5 (6) | 14 (15) |
| | Off-street dedicated routes | 10 (6) | 6 (7) | 4 (4) |
| | Protected routes | 14 (8) | 8 (9) | 6 (6) |
| | Unspecified routes | 59 (33) | 30 (35) | 29 (31) |
| | Maintenance | 5 (3) | 1 (1) | 4 (4) |
| | Connectivity | 22 (12) | 10 (12) | 12 (13) |
| | Convenience | 7 (4) | 3 (3) | 4 (4) |
| | 7th Avenue bicycle boulevard | 30 (17) | 10 (12) | 20 (22) |
| | Infrastructure safety | 4 (2) | 1 (1) | 3 (3) |
| Road infrastructure | Roundabouts | 4 (2) | 2 (2) | 2 (2) |
| | Intersections/crosswalks | 6 (3) | 2 (2) | 4 (4) |
| | Signage | 13 (7) | 7 (8) | 6 (6) |
| | Maintenance | 2 (1) | 0 (0) | 2 (2) |
| Promotion and education | Sharing the road | 68 (38) | 35 (41) | 33 (35) |
| | Awareness of infrastructure | 14 (8) | 8 (9) | 6 (6) |
| | Use of safety equipment | 5 (3) | 2 (2) | 3 (3) |
| | Active transportation promotion | 5 (3) | 3 (3) | 2 (2) |
| Policy | Economics | 6 (3) | 3 (3) | 3 (3) |
| | Laws | 3 (2) | 1 (1) | 2 (2) |
| | Enforcement | 6 (3) | 2 (2) | 4 (4) |
| Public perceptions and attitudes | Norms | 9 (5) | 4 (5) | 5 (5) |
| | Respect | 5 (3) | 3 (3) | 2 (2) |
| | Safety | 40 (22) | 25 (29) | 15 (16) |
| | Other | 19 (11) | 8 (9) | 11 (12) |

Of note, 30 participants (10 women and 20 men) commented specifically about the 7th Avenue bicycle boulevard and 73.3% of these comments were positive (60% for women and 80% for men). For example, a 28-year-old man who reported using a mix of cycling, walking, and driving for transportation mentioned: "*Well, I like what they have done with 7th here. They have totally traffic-calmed it and it's way nicer to ride on. So, more bike routes... I'm from Vancouver originally and we got bike routes criss-crossing the entire city. There is basically that one [bike boulevard] going that way but there is none going North–South.*" As in this quote, many participants advocated for a better-connected network of cycling infrastructure. Other participants advocated for more bicycle boulevards in general. For example, a 23-year-old woman who used a combination of cycling, walking, and driving said: "*Uh, I just think more of these bike boulevards would be a great idea. It's less nerve-racking riding down a simple street like this, rather than riding in the city streets being around all the cars and what not, it's just way more easygoing and fun.*"

Participants who expressed negative views towards the bicycle boulevard generally criticized the cost or were concerned about drivers' lack of understanding of how to navigate novel infrastructure like diverters and mini roundabouts. A 68-year-old man who reported using a mix of cycling, walking, and driving for transportation said: "*7th Ave is a waste of money, 2.7 million total waste of money. I don't know who was responsible for it. No one's taking the blame for it, or responsibility for it.*" A 63-year-old woman who used a combination of cycling and walking for transportation mentioned: "*I would prefer that the traffic circles didn't exist. Drivers don't know how to use them. I have been nearly hit more on the 7th Avenue corridor than anywhere else in my life.*" Some of these negative views appeared in line with negative stories published in local media [30–33].

## 4. Discussion

This mixed-methods study examined the relationships between the presence of cycling infrastructure, psychosocial and socio-demographic factors, and cycling for transportation in a small city that had a low prevalence of cycling to work according to census data. Participants reported an increase in cycling time over a time period when new infrastructure was introduced. Although men reported significantly more cycling time than women, women intercepted on the street with a separate multi-use path cycled more than those intercepted on the street with no cycling infrastructure. Safety concerns were higher among participants interviewed on the street with no cycling infrastructure. In open-ended questions, most people expressed favorable views of the bicycle boulevard and the majority of participants identified a need for better cycling infrastructure in general. Taken together, our results suggest that investments in cycling infrastructure have the potential to positively impact cycling behaviour, even in small communities with a low prevalence of transportation cycling.

Our multivariable models (Table 2) illustrate that women surveyed along the segregated multi-use path reported at least twice as much time spent cycling compared to those interviewed along the street without cycling infrastructure. This observation is consistent with our qualitative data and previous literature suggesting that people prefer cycling on dedicated routes or traffic-calmed streets [14,18,19,22]. In contrast, we observed that men interviewed along the bicycle boulevard reported less cycling than those interviewed along the street with no cycling infrastructure. In a previous study in Portland, Oregon, Broach et al. [18] found that women were more willing than men to make a detour to use bicycle boulevards. This suggests that men may prefer more direct routes, and future research is needed to explore this possibility. For instance, future studies on cycling infrastructure in small cities could investigate actual routes traveled by men and women with global positioning systems (GPS) and examine how they consider trade-offs between route directedness and safety.

### 4.1. Sociodemographic Correlates of Cycling Time

The observation that men cycled more than women is consistent with previous research in predominantly English-speaking countries such as Australia, Canada, the UK, and the US [7,15,44]. Interestingly, in countries where cycling is safer and more prevalent, such as the Netherlands, Denmark, and Germany, there are no gender differences in transportation cycling [15]. Researchers studying gender differences in cycling have also argued that women represent "indicator species" for cycling-friendly cities [48].

We observed that time spent cycling decreased by 2–3% for each additional year of age in women and increased by 2% with each year in men. Some previous studies found that cycling declines with age [7,15,44] and others found no effect [49]. Given the nature of intercept interviews, it is possible that our older interviewees were more enthusiastic about cycling than older adults in general. We also noted that women with college or university education cycled more than those who held a high school diploma or less. However, men with college education cycled less than those with a lower level of education. Previous research examining the association between level of education remains inconclusive [7,44,50,51]. Lastly, we found no association between vehicle ownership and reported cycling time in multivariable models. Previous literature suggests that car owners are less likely to cycle to work [52]. We suspect that vehicle ownership would be more strongly associated with travel mode than the amount of time spent cycling, and most previous studies have focused on travel mode.

### 4.2. Psychological Correlates of Cycling Time

Attitudes towards cycling were the most consistent correlate of cycling in our models. For each unit increase in a scale ranging from 1 to 5, cycling time increased by about 2 to 5 times. In their survey of cycling in 6 US cities, Emond et al. [50] also reported that attitudes were one of few correlates that was not gender-specific. The important role of

attitudes is consistent with the broader literature on cycling for transportation [23,43,53,54]. Furthermore, attitudes are a central concept in theories such as the theory of planned behaviour [55] and the theory of interpersonal behaviour [56]. These observations suggest that theory-based interventions and social marketing campaigns to improve attitudes towards cycling may be promising. We also found that a more favorable attitude towards walking was associated with lower cycling time. Based on the attitudinal items used (Supplementary File S1), our findings suggest that individuals who prefer walking over other travel modes cycle less.

We found that habit strength was significantly associated with cycling time in men and women, which is consistent with previous work highlighting the importance of habits as a determinant of travel mode [38,54,57,58]. According to the theory of interpersonal behaviour, as a habit gets stronger, the relationship between attitudes and behaviour diminishes [56]. In other words, executing habitual behaviours in response to cues (e.g., the need to travel to work) can become automatic and no longer requires conscious deliberation about different travel modes [58]. Conversely, the habit discontinuity hypothesis suggests that events that change contextual cues such as home or workplace relocation can be opportunistic times to deliver interventions that promote behaviour change [59,60].

We observed that men who had greater concerns about traffic in their neighbourhoods reported more cycling time. Among women, there was no association between safety concerns and cycling time. Although our findings appear counterintuitive, they may suggest that cyclists are more aware of traffic volume and speed in their neighbourhood based on the items included in our survey (see Supplementary File S1). Alternatively, traffic concerns may be more related to whether individuals cycle or not rather than to the amount of time they spend cycling. Previous studies indicate that road safety concerns are an important deterrent to cycling [43,44,54,61,62]. For instance, Sallis et al. [44] observed that half of individuals who never cycled would consider riding if safety improved. Our qualitative data also suggested that perceived safety was a concern for many respondents.

### 4.3. Cycling Infrastructure and Participants' Suggestions to Make Cycling Safer

The need for better infrastructure for transportation and/or leisure cycling was the most frequent theme identified in participants' qualitative responses. This was interesting given some of the negative attention that had been directed to the municipal government for investing in the bicycle boulevard. Some interviewees who were less supportive of the bicycle boulevard believed that it cost substantially more than it did: according to the city, it cost $595,000 rather than $2.7 million. This emphasizes the need for clear communication, which may be particularly important in smaller communities where cycling infrastructure is still novel and more contested.

Participants' stated preferences for separated paths and/or on-road infrastructure is consistent with previous studies [18,19,22]. Our findings suggests that even in smaller cities, cycling infrastructure designed to increase safety is desirable and that greater separation from traffic may encourage women to cycle more. Many participants, especially women, expressed concerns about safety and emphasized the need for education of drivers and/or cyclists about the rules of the road, which is consistent with previous qualitative research in larger metropolitan areas [61,63]. Many participants also emphasized the need for increased connectivity in the cycling network, and previous research appears to support this view [39,62]. It is worth noting that a link between the bicycle boulevard and a major north–south path was still under construction during data collection (Figure 1). The segregated path was linked with the same north–south path through an underpass that avoided an interaction with a busy arterial road. This may partly explain why women interviewed along that path reported more cycling.

### 4.4. Strengths and Limitations

Study strengths include the mixed methods design, the use of quantitative measures with documented reliability and validity, and the involvement of multiple researchers in

coding qualitative data. In contrast, the fact that changes in cycling time were examined retrospectively makes it impossible to determine that the construction of cycling infrastructure caused an increase in cycling. Furthermore, travel behaviours and potential correlates of cycling were assessed by self-report and are subject to social desirability and recall bias. The generalizability of findings to other cities is unclear, underscoring a need for future studies of bicycle boulevards in small cities. Despite our efforts to recruit a diverse sample, intercept surveys are vulnerable to selection bias and the values for time spent cycling were higher than expected, though comparable with an online survey in Australia [25]. Given our sampling strategy, changes in cycling between 2016 and 2018 may be larger than they would have been if we had recruited a random sample of adults in Lethbridge. This limitation may have been reduced by the greater ease of intercepting walkers compared to fast moving cyclists.

## 5. Conclusions

Our findings suggest that cycling infrastructure has the potential to favorably impact perceived safety and cycling behaviour, especially among women. Participants reported that they cycled significantly more than they did 2 years prior, before the construction of the bicycle boulevard. Qualitative findings also suggested generally positive perceptions of the bicycle boulevard and a desire for more cycling infrastructure. These results provide preliminary evidence that investments in cycling infrastructure may benefit smaller communities with a low prevalence of active transportation. Policy-makers at the municipal, provincial, and federal level can contribute to funding such infrastructure. Our qualitative results also provide locally-relevant insights. Local authorities should work with relevant stakeholders (e.g., cycling associations, communication specialists, municipal police, etc.) on education campaigns about sharing the road. Local authorities should also improve the connectivity of the cycling network to reduce safety concerns and increase transportation cycling. Given the well-established environmental and public health benefits of transportation cycling, longer and larger prospective studies of travel behaviour change are warranted. Researchers should work in collaboration with urban planners and transportation engineers to ensure the possibility of collecting data before and after the implementation of new infrastructure [64].

**Supplementary Materials:** The following are available online at https://www.mdpi.com/article/10.3390/futuretransp1010007/s1, Supplementary File S1: Transportation Questionnaire. Supplementary File S2: Bicycle Boulevard Map, Part 1. Supplementary File S3: Bicycle Boulevard Map, Part 2.

**Author Contributions:** Conceptualization, R.L., J.L.C., and N.P.; methodology, R.L., J.L.C., and N.P.; software, R.L.; validation, R.L.; formal analysis, R.L.; investigation, R.L.; resources, R.L. and J.L.C.; data curation, R.L.; writing—original draft preparation, R.L.; writing—review and editing, J.L.C. and N.P.; supervision, R.L.; project administration, R.L.; funding acquisition, R.L. All authors have read and agreed to the published version of the manuscript.

**Funding:** This research was funded by the University of Lethbridge, fund number 14450.

**Institutional Review Board Statement:** The study was conducted according to the guidelines of the Declaration of Helsinki, and approved by the Institutional Review Board (or Ethics Committee) of University of Lethbridge (protocol code 2018-036, approved 12 April 2018).

**Informed Consent Statement:** Informed consent was obtained from all subjects involved in the study.

**Data Availability Statement:** The data presented in this study are available on request from the corresponding author via email.

**Acknowledgments:** The authors acknowledge the help of Sahra Nodge, Ali Walker, Kiana Althen, Reilly Parkinson, Tyler Priest, and Tyson Atkinson in data collection and/or qualitative data analysis. The authors also acknowledge the contribution of the Transportation Engineering department at the City of Lethbridge in identifying suitable control streets.

**Conflicts of Interest:** The first author receives book royalties from Elsevier. The other authors have no potential conflicts of interest to declare.

## Appendix A

**Table A1.** Relationship between type of infrastructure, gender, and time spent cycling.

| Variable | Exp (β) | 95% CI |
|---|---|---|
| Street—segregated path (ref: no infrastructure) | 1.74 | 1.03–2.96 * |
| Street—bicycle boulevard (ref: no infrastructure) | 1.76 | 1.08–2.85 * |
| Gender–men (ref: women) | 4.84 | 2.78–8.46 ** |
| Gender * Street—segregated path (ref: no infrastructure and women) | 0.47 | 0.23–0.98 * |
| Gender * Street—bicycle boulevard (ref: no infrastructure and women) | 0.42 | 0.22–0.82* |

Note: * indicates statistical significance ($p < 0.05$); ** indicates $p < 0.001$. Akaike Information Criteria: 3035.690.

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
