# Peer review of "Relationship between Cycling Infrastructure and Transportation Cycling in a Small Urban Area"

_futuretransp, doi:10.3390/futuretransp1010007_

Round 1
Reviewer 1 Report
Page 1, Title of the paper, “Relationship between cycling infrastructure and transportation 2 cycling in a small urban area: a pilot study”: My suggestion is to change the title to “Relationship between cycling infrastructure and transportation cycling in a small urban area”. I do not think that “a pilot study” add something more to the title.
Page 1, Abstract: Please add the name of the small urban area (Lethbridge, Alberta) in the abstract since it will be useful for the reader to understand from the early beginning of the paper which is your case study.
Section 2. Materials and Methods: My suggestion is to include a Data Flow Diagram including all your methodological steps so to allow the reader to obtain a clear overview of your work.
Section 2. Materials and Methods, subsection 2.2 Participants: Please provide full details concerning the sampling method used in your questionnaire-based survey. Which is the sample size calculation formula used?
My suggestion is to include a map and some photographs of the study area (e.g., the bicycle boulevard, the control streets etc.).
Please try to enhance the policy recommendations included in the “Conclusions”. Please also try to associate each one of the policy recommendations with the respective stakeholder (e.g, municipal authority, cycling associations etc.).
Author Response
Thank you very much for your timely review. Please find attached our responses to the comment from both reviewers.

Reviewer 2 Report
The authors present an interesting study investigating perceptions towards cycling infrastructure in a Lethbridge Alberta, Canada. The manuscript is written well. The procedure is easy to follow and the methods are well documented. Overall I feel the authors have presented an interesting mixed methods study that would be of interest to readers.
I have a few minor comments that I would like to see the author address prior to publication.
Line 108: Please include the street names of the control sites.
Line 119: Reword this sentence it is unclear if dogwalkers and people sitting in the park were included in the sample.
Line 175: Reword this sentence it is poorly constructed.
Line 186: I would like to see the model included in the body of the manuscript. It would be of interest to the reader. At the very least it should be presented in an appendix. This also applies to the data discussed on line 216.
Line 188: Self-citation of the lead authors previous analysis is not strong justification for the methodology. Revise this section.
Author Response

(The authors gave the same response as above.)
